# Label Words are Anchors: An Information Flow Perspective for Understanding In-Context Learning

**Lean Wang[†,§], Lei Li[†], Damai Dai[†], Deli Chen[§],**
**Hao Zhou[§], Fandong Meng[§], Jie Zhou[§], Xu Sun[†]**
[†]National Key Laboratory for Multimedia Information Processing,
School of Computer Science, Peking University
[§]Pattern Recognition Center, WeChat AI, Tencent Inc., China
{lean,daidamai,xusun}@pku.edu.cn  nlp.lilei@gmail.com
victorchen@deepseek.com  {tuxzhou,fandongmeng,withtomzhou}@tencent.com

## Abstract

In-context learning (ICL) emerges as a promising capability of large language models (LLMs) by providing them with demonstration examples to perform diverse tasks. However, the underlying mechanism of how LLMs learn from the provided context remains under-explored. In this paper, we investigate the working mechanism of ICL through an information flow lens. Our findings reveal that label words in the demonstration examples function as anchors: (1) semantic information aggregates into label word representations during the shallow computation layers' processing; (2) the consolidated information in label words serves as a reference for LLMs' final predictions. Based on these insights, we introduce an anchor re-weighting method to improve ICL performance, a demonstration compression technique to expedite inference, and an analysis framework for diagnosing ICL errors in GPT2-XL. The promising applications of our findings again validate the uncovered ICL working mechanism and pave the way for future studies.[1]

## 1 Introduction

In-context Learning (ICL) has emerged as a powerful capability alongside the development of scaled-up large language models (LLMs) (Brown et al., 2020). By instructing LLMs using few-shot demonstration examples, ICL enables them to perform a wide range of tasks, such as text classification (Min et al., 2022a) and mathematical reasoning (Wei et al., 2022). Since ICL does not require updates to millions or trillions of model parameters and relies on human-understandable natural language instructions (Dong et al., 2023), it has become a promising approach for harnessing the full potentiality of LLMs. Despite its significance, the inner working mechanism of ICL remains an open question, garnering considerable interest from research

[1]https://github.com/lancopku/label-words-are-anchors

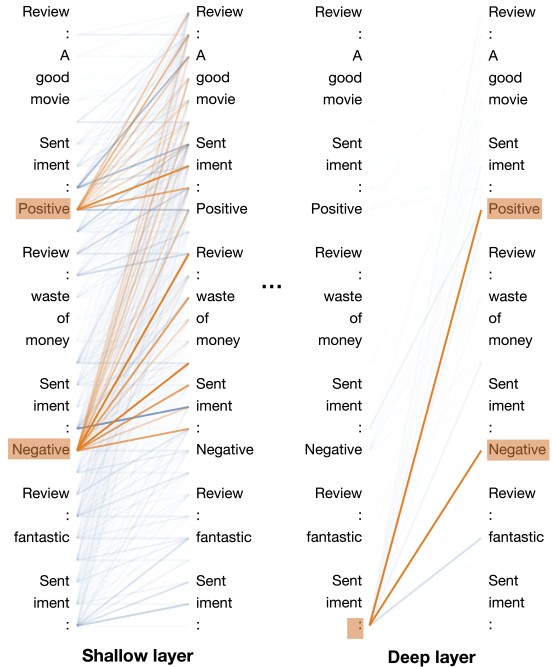

Figure 1: Visualization of the information flow in a GPT model performing ICL. The line depth reflects the significance of the information flow from the right word to the left. The flows involving label words are highlighted. Label words gather information from demonstrations in shallow layers, which is then extracted in deep layers for final prediction.

communities (Xie et al., 2022; Dai et al., 2022; Akyürek et al., 2022; Li et al., 2023b).

In this paper, we find that the label words serve as anchors that aggregate and distribute information in ICL. We first visualize the attention interactive pattern between tokens with a GPT model (Brown et al., 2020) on sentiment analysis (Figure 1). Initial observations suggest that label words aggregate information in shallow layers and distribute it in deep layers.[2] To draw a clearer picture of this phenomenon, we design two metrics based on saliency

---

[2]In this paper, "shallow" or "first" layers refer to those closer to the input, while "deep" or "last" layers are closer to the output. Here, "deep layers" include those around the midpoint, e.g., layers 25-48 in a 48-layer GPT2-XL.

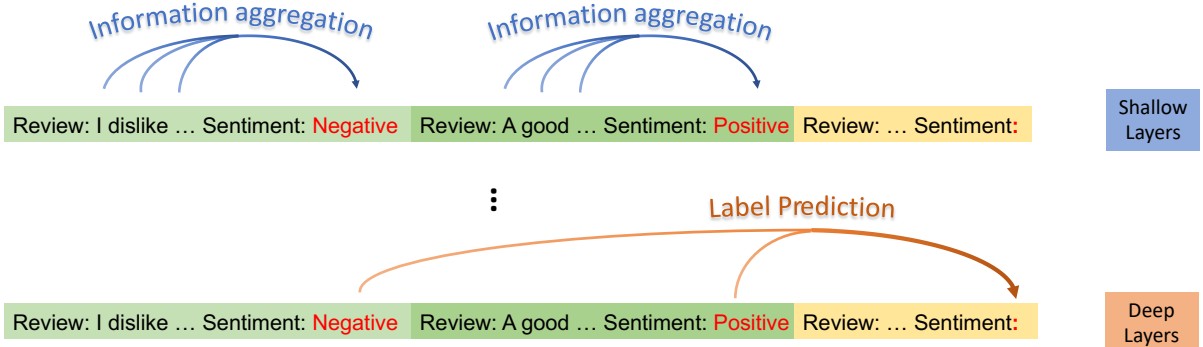

Figure 2: Illustration of our hypothesis. In shallow layers, label words gather information from demonstrations to form semantic representations for deeper processing, while deep layers extract and utilize this information from label words to formulate the final prediction.

scores to portray the information flow in ICL and further propose the following hypothesis:

> *Information Flow with Labels as Anchors*
> $\mathcal{H}_1$: In shallow layers, label words gather the information of demonstrations to form semantic representations for deeper layers.
> $\mathcal{H}_2$: In deep layers, the model extracts the information from label words to form the final prediction.

Two experiments are designed to validate the hypothesis using GPT2-XL (Radford et al., 2019) and GPT-J (Wang and Komatsuzaki, 2021) across several text classification benchmarks. (1) By blocking the information aggregation path to label words in certain layers, we find that such isolation in shallow layers significantly impairs model performance. This indicates that label words collect useful information during forward propagation in shallow layers. (2) We investigate the relationship between the attention distributions on the label words of the target position and the model's final prediction. Our results illustrate a strong positive correlation, where a candidate label's probability increases with more attention weight on its corresponding label token. In summary, these experimental findings suggest that our hypothesis holds well with large language models on real-world datasets.

Drawing on insights from the information flow perspective, we explore three approaches to enhance ICL's effectiveness, efficiency, and interpretability. (1) An anchor re-weighting method is introduced, which employs a learnable vector to adjust the significance of different label words in demonstrations, leading to a 16.7% average accuracy boost compared to standard ICL baselines. (2) For quicker ICL inference, inputs are compressed into pre-calculated anchor representations since model predictions primarily rely on label word activations. Testing shows a $1.8 \times$ speedup in inference with only a minimal performance trade-off. (3) An error analysis of ICL on GPT2-XL demonstrates that the label confusion matrix aligns closely with the distance distribution of anchor key vectors, implying that errors might result from similar anchor representations. These promising applications further validate our hypothesis and shed light on future ICL studies for better transparency of LLMs.

## 2 Label Words are Anchors

This section confirms the intuitive findings using two saliency score-based metrics as discussed in § 2.1. The quantitative results lead to a proposed hypothesis for the ICL working mechanism: $\mathcal{H}_1$: In shallow layers, label words aggregate information from demonstration examples to form semantic representations for later computations. $\mathcal{H}_2$: In deep layers, the model makes predictions by extracting information from label words. The validation for these hypotheses is presented in § 2.2 and § 2.3, respectively.

### 2.1 Hypothesis Motivated by Saliency Scores

This section aims to discover the inherent patterns in the attention interaction between tokens for a GPT model. The saliency technique (Simonyan et al., 2013), a common interpretation tool, is employed for highlighting critical token interactions. Following common practice, we use the Taylor expansion (Michel et al., 2019) to calculate the saliency score for each element of the attention matrix:

$$I_l = \sum_h \left| A_{h,l}^\top \frac{\partial \mathcal{L}(x)}{\partial A_{h,l}} \right|. \tag{1}$$

Here, $A_{h,l}$ is the value of the attention matrix of the $h$-th attention head in the $l$-th layer, $x$ is the input, and $\mathcal{L}(x)$ is the loss function of the task, e.g., the cross-entropy objective for a classification problem. We average all attention heads to obtain the saliency matrix $I_l$ for the $l$-th layer. $I_l(i,j)$ represents the significance of the information flow from the $j$-th word to the $i$-th word for ICL. By observing $I_l$, we can get an intuitive impression that as the layer goes deeper, demonstration label words will become more dominant for the prediction, as depicted in Figure 1.

To draw a clearer picture of this phenomenon, we propose three quantitative metrics based on $I_l$. Our focus lies in three components: (i) the label words, such as "Negative" and "Positive" in Figure 2, denoted as $p_1, ..., p_C$, where $C$ represents the total number of label words;[3] (ii) the target position, where the model generates prediction labels (i.e., the final token in the input), which we denote as $q$; and (iii) the text part, i.e., the tokens before label words in the demonstration.

The definitions of the three quantitative metrics follow below.

$S_{wp}$, the mean significance of information flow from the text part to label words:

$$S_{wp} = \frac{\sum_{(i,j) \in C_{wp}} I_l(i,j)}{|C_{wp}|},$$
$$C_{wp} = \{(p_k, j) : k \in [1, C], j < p_k\}. \quad (2)$$

$S_{pq}$, the mean significance of information flow from label words to the target position:

$$S_{pq} = \frac{\sum_{(i,j) \in C_{pq}} I_l(i,j)}{|C_{pq}|},$$
$$C_{pq} = \{(q, p_k) : k \in [1, C]\}. \quad (3)$$

$S_{ww}$, the mean significance of the information flow amongst all words, excluding influences represented by $S_{wp}$ and $S_{pq}$ :

$$S_{ww} = \frac{\sum_{(i,j) \in C_{ww}} I_l(i,j)}{|C_{ww}|},$$
$$C_{ww} = \{(i,j) : j < i\} - C_{wp} - C_{pq}. \quad (4)$$

$S_{wp}$, $S_{pq}$, and $S_{ww}$ help assess different information flows in the model. $S_{wp}$ indicates the intensity of information aggregation onto label words. A high $S_{pq}$ demonstrates a strong information extraction from label words for final decision-making.

---

[3]In this study, the term 'label words' is approximately equal to 'label tokens'. The only deviation is the 'Abbreviation' in the TREC dataset, where we use the first subword in experiments, following Zhao et al. (2021).

$S_{ww}$ assesses average information flow among words, serving as a benchmark to gauge the intensity of the patterns identified by $S_{wp}$ and $S_{pq}$.

**Experimental Settings** We choose GPT2-XL from the GPT series (Radford et al., 2019) as our primary model for investigation, due to its moderate model size (of 1.5B parameters) that is suitable for our hardware resource and its decent ICL performance (Dai et al., 2022). For datasets, we use Stanford Sentiment Treebank Binary (SST-2) (Socher et al., 2013) for sentiment analysis, Text REtrieval Conference Question Classification (TREC) (Li and Roth, 2002; Hovy et al., 2001) for question type classification, AG's news topic classification dataset (AGNews) (Zhang et al., 2015) for topic classification, and EmoContext (EmoC) (Chatterjee et al., 2019) for emotion classification. Templates for constructing demonstrations are provided in Appendix A. 1000 examples are sampled from the test set for evaluation, with one demonstration per class sampled from the training set. Experiments with more demonstrations yield similar outcomes (refer to Appendix F.1 for details). Results reflect averages from five random seeds.

**Results and Analysis** Figure 3 reveals that: (1) in shallow layers, $S_{pq}$, the significance of the information flow from label words to targeted positions, is low, while $S_{wp}$, the information flow from the text part to label words is high; (2) in deep layers, $S_{pq}$, the importance of information flow from label words to the targeted position becomes the dominant one. Notably, $S_{pq}$ and $S_{wp}$ usually surpass $S_{ww}$, suggesting that interactions involving label words outweigh others.

**Proposed Hypothesis** Based on this, we propose the hypothesis that label words function as anchors in the ICL information flow. In shallow layers, label words gather information from demonstration examples to form semantic representations for deeper layers, while in deep layers, the model extracts the information from label words to form the final prediction. Figure 2 gives an illustration for our hypothesis.

## 2.2 Shallow Layers: Information Aggregation

In this part, we validate our hypothesis' first component. We assume that the information aggregation in ICL relies on the information flow from the text part to label tokens, which is facilitated by the transformer's attention mechanism. By manipulating

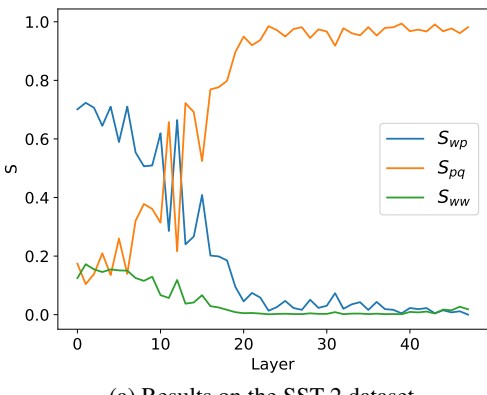

(a) Results on the SST-2 dataset

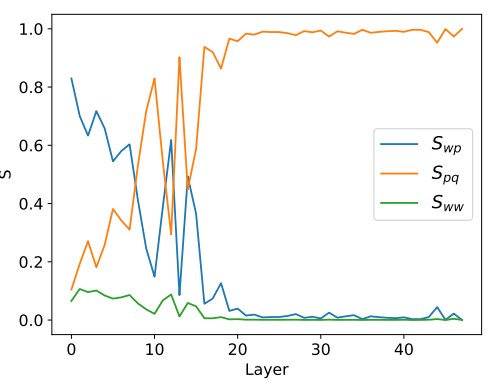

(b) Results on the AGNews dataset

Figure 3: Relative sizes of $S_{wp}$, $S_{pq}$, and $S_{ww}$ in different layers on SST-2 and AGNews. Results of other datasets can be found in Apendix B. Initially, $S_{wp}$ occupies a significant proportion, but it gradually decays over layers, while $S_{pq}$ becomes the dominant one.

the attention layer in the model to block this flow and examining the model behavior change, we validate the existence of the information aggregation process and its contribution to the final prediction.

**Experimental Settings** We retain the same test sample size of 1000 inputs as § 2.1. We use the same demonstration for a single random seed. To further validate our findings on larger models, we incorporate GPT-J (6B) (Wang and Komatsuzaki, 2021) in experiments, which exceeds GPT2-XL in model size and capacity.

**Implementation Details** To block the information flow to label words, we isolate label words by manipulating the attention matrix $A$. Specifically, we set $A_l(p,i)(i < p)$ to 0 in the attention matrix $A_l$ of the $l$-th layer, where $p$ represents label words and $i$ represents preceding words. Consequently, in the $l$-th layer, label words cannot access

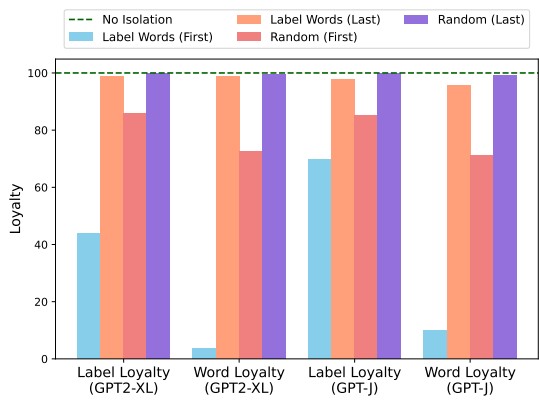

Figure 4: The impact of isolating label words versus randomly isolating non-label words within the first or last 5 layers. Isolating label words within the first 5 layers exerts the most substantial impact, highlighting the importance of shallow-layer information aggregation via label words.

information from the prior demonstration text.

**Metrics** We use the following metrics to assess the impact of blocking information flow from the text part to label tokens: **(1) Label Loyalty:** measures the consistency of output labels with and without isolation. **(2) Word Loyalty:** employs the Jaccard similarity to compare the top-5 predicted words with and without isolation, capturing more subtle model output alterations (See Appendix C for details). Low loyalty indicates a profound impact of isolation on model predictions.

**Results and Analysis** Figure 4 illustrates a notable influence on the model's behavior when label words are isolated within the first 5 layers. Yet, this influence becomes inconsequential within the last 5 layers, or when random non-label words are used. This observation underlines the fundamental importance of shallow-layer information aggregation via label words in ICL. It also emphasizes the superiority of label words over non-label words. Further tests with variable numbers of layers reaffirm these findings (Appendix D). Moreover, similar results were obtained when testing ICL with semantically unrelated labels (refer to Appendix F.2).

## 2.3 Deep Layers: Information Extraction

We proceed to validate the latter part of our hypothesis that the model extracts information from label words to form the final prediction. We denote the sum of the attention matrices in the $l$-th

layer as $A_l$.[4] In deeper layers, we find a strong correlation between the attention distributions on the label words of the target position, represented as $(A_l(q, p_1), ..., A_l(q, p_C))$, and the model's final prediction, affirming our hypothesis. The experimental setup mirrors that discussed in § 2.2.

### 2.3.1 Experiments

We utilize the AUC-ROC score to quantify the correlation between $A_l(q, p_i)$ and model prediction, which we denote as $\text{AUCROC}_l$ for the $l$-th layer. We prefer the AUC-ROC metric due to two primary reasons: (1) $A_l(q, p_i)$ might differ from the probability of the model outputting label $i$ by a constant factor. As Kobayashi et al. (2020) points out, attention should be multiplied by the norm of the key vector to yield 'more interpretable attention'. The AUC-ROC metric can implicitly account for these factors, thus allowing us to uncover the correlation more effectively. (2) The proportion of different labels output by the model may be unbalanced. Using the AUC-ROC metric can help mitigate this issue, reducing disturbances caused by class imbalance.

Considering the residual mechanism of transformers, we can view each layer's hidden state as the cumulative effect of all prior layer calculations. To quantify the accumulated contribution of the first $l$ layers to model prediction, we introduce $R_l$:

$$R_l = \frac{\sum_{i=1}^{l}(\text{AUCROC}_i - 0.5)}{\sum_{i=1}^{N}(\text{AUCROC}_i - 0.5)}. \quad (5)$$

This measure tracks the positive contribution above a baseline AUC-ROC threshold of 0.5. The value of $R_l$ signifies the proportional contribution of the first $l$ layers to the model prediction.

### 2.3.2 Results and Analysis

Figures 5a and 5b delineate correlation metrics for GPT2-XL and GPT-J, averaged across four datasets. The $\text{AUCROC}_l$ for deep layers approaches 0.8, illustrating a strong correlation between the attention distributions on label words of the target position and the model's final prediction. Moreover, shallow layers show negligible cumulative contributions ($R_l$), with a significant increase in middle and deep layers. These results signify the crucial role of deep layers for final prediction, validating that the model extracts information from label words in deep layers to form the final prediction.

---

[4]Here we sum up the attention matrices of all attention heads in the $l$th layer for convenience of analysis.

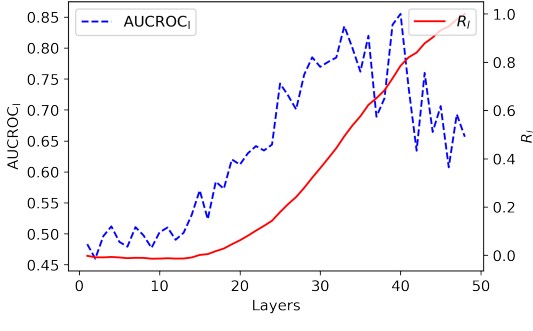

(a) GPT2-XL (total 48 layers).

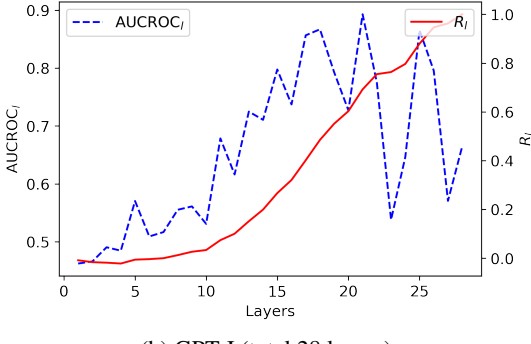

(b) GPT-J (total 28 layers).

Figure 5: $\text{AUCROC}_l$ and $R_l$ of each layer in GPT models. The result is averaged over SST-2, TREC, AGNews, and Emoc. $\text{AUCROC}_l$ reaches 0.8 in deep layers, and $R_l$ increases mainly in the middle and later layers.

### 2.4 Discussion of Our Hypothesis

In § 2.2, we have affirmed that the model's shallow layers assemble information from demonstrations via label words to form semantic representations. In § 2.3, we verify that the aforementioned aggregated information on label words is then extracted to form the final prediction in the deep layers. Recognizing the crucial function of label words in this process, we have introduced the term "Anchors" to denote them. Given the considerable role these "anchors" fulfill, we find it intuitive to design ICL improvements based on them, as elaborated in § 3.

## 3 Applications of Our Anchor-Based Understanding

With insights from the validated hypothesis, we propose strategies to boost ICL's accuracy and inference speed. We propose an anchor re-weighting method in § 3.1 to adjust the demonstrations' contributions and improve accuracy. In § 3.2, we explore a context compression technique that reduces original demonstrations to anchor hidden states to speed up ICL inference. Besides, in § 3.3, we utilize anchor distances to perform an analysis to understand

the errors ICL made in real-world scenarios. These approaches corroborate our hypothesis, pointing to potential paths for future ICL enhancements.

## 3.1 Anchor Re-weighting

Based on our analysis in § 2, we draw parallels between ICL and logistic regression and propose an approach to improve ICL's accuracy by re-weighting label anchors.

### 3.1.1 Method

§ 2.3 illustrates a strong correlation between the model's output category and the attention distribution $(A(q, p_1), \ldots, A(q, p_C))$ on label words $p_1, \ldots, p_C$ of the target position $q$ in deep layers. We can view the attention module as a classifier $\boldsymbol{f}$,

$$
\begin{aligned}
&\Pr_{\boldsymbol{f}}(Y = i | X = x) \\
&\approx A(q, p_i) \\
&= \frac{\exp(\mathbf{q}_q \mathbf{k}_{p_i}^T / \sqrt{d})}{\sum_{j=1}^{N} \exp(\mathbf{q}_q \mathbf{k}_j^T / \sqrt{d})}.
\end{aligned} \tag{6}
$$

By setting $\mathbf{q}_q / \sqrt{d} = \hat{\mathbf{x}}$ and $\mathbf{k}_{p_i} - \mathbf{k}_{p_C} = \boldsymbol{\beta}_i$, we deduce:

$$
\log \frac{\Pr_{\boldsymbol{f}}(Y = i | X = x)}{\Pr_{\boldsymbol{f}}(Y = C | X = x)} = \boldsymbol{\beta}_i^T \hat{\mathbf{x}}. \tag{7}
$$

This approximates a logistic regression model where:

$$
\log \frac{\Pr_{\boldsymbol{f}}(Y = i | X = x)}{\Pr_{\boldsymbol{f}}(Y = C | X = x)} = \beta_0^i + \boldsymbol{\beta}_i^T \mathbf{x}. \tag{8}
$$

In this equation, $\beta_0^i$ and $\boldsymbol{\beta}_i^T$ are parameters that can be learned, while $\mathbf{x}$ is the input feature.

Inspired by the similarity between ICL and logistic regression, we've incorporated a learnable $\beta_0^i$ into Eq. (7), which is equivalent to adjusting the attention weights $A(q, p_i)$:

$$
\hat{A}(q, p_i) = \exp(\beta_0^i) A(q, p_i) \tag{9}
$$

Each $\beta_0^i$ is a learnable parameter, set uniquely for different attention heads and layers. Refer to Appendix G for more details.

To train the re-weighting vector $\boldsymbol{\beta} = \{\beta_0^i\}$, we utilize an auxiliary training set $(\boldsymbol{X}_{train}, \boldsymbol{Y}_{train})$. Here, we perform ICL with normal demonstrations and optimize $\boldsymbol{\beta}$ with respect to the classification loss $\mathcal{L}$ on $(\boldsymbol{X}_{train}, \boldsymbol{Y}_{train})$:

$$
\boldsymbol{\beta}^\star = \arg \min_{\boldsymbol{\beta}} \mathcal{L}(\boldsymbol{X}_{train}, \boldsymbol{Y}_{train}). \tag{10}
$$

This approach can be metaphorically described as "re-weighting the anchors," leading us to term it

as **Anchor Re-weighting**. It can also be viewed as a modification of the demonstration contributions since demonstration information has been incorporated into the anchors as suggested by our prior analysis in § 2.2. Additionally, it can be interpreted as a unique adapter variant, introducing minimal parameters while preserving most of the original model. However, it is specifically designed based on our anchor hypothesis and requires fewer parameters than traditional adapters.

### 3.1.2 Experiments

We choose one sample per class as normal demonstrations and choose four extra samples per class to form the auxiliary training set $(\boldsymbol{X}_{train}, \boldsymbol{Y}_{train})$. The setup follows § 2.2, with results averaged over five random seeds. Owing to computational constraints, we employ GPT2-XL for evaluation, excluding GPT-J. The parameters $\{\beta_0^i\}$ are trained using gradient descent. More details can be found in Appendix H.

We compare **Anchoring Re-weighting** with two baselines: (1) Vanilla ICL with the same demonstration (1-shot per class) (2) Vanilla ICL, where the auxiliary training set of $\boldsymbol{\beta}$ is included as demonstrations (5-shot per class) for a fair comparison.

### 3.1.3 Results

As Table 1 shows, the proposed anchor re-weighting significantly enhances ICL performance, particularly on the SST-2 and EmoC datasets. Besides, adding more demonstrations for vanilla ICL may not bring a stable accuracy boost due to the potential noise introduced, as discussed in Zhao et al. (2021). Different from vanilla ICL which utilizes the extra examples to form a demonstration, we train a re-weighting vector $\boldsymbol{\beta}$ to modulate label anchor contributions. This shortens the input context and thus brings (almost) no extra cost to the inference speed. The consistent improvements of our method suggest that the re-weighting mechanism could be a better alternative to utilize demonstration examples. Furthermore, it reiterates the crucial role that anchors play in ICL.

## 3.2 Anchor-Only Context Compression

We further explore a context compression technique that reduces the full demonstration to anchor hidden states for accelerating ICL inference.

### 3.2.1 Method

In § 2.3, we find that the model output heavily relies on the label words, which collect information

| Method | SST-2 | TREC | AGNews | EmoC | Average |
|---|---|---|---|---|---|
| Vanilla In-Context Learning ( 1-shot per class ) | 61.28 | 57.56 | 73.32 | 15.44 | 51.90 |
| Vanilla In-Context Learning ( 5-shot per class ) | 64.75 | 60.40 | 52.52 | 9.80 | 46.87 |
| Anchor Re-weighting (1-shot per class) | **90.07** | **60.92** | **81.94** | **41.64** | **68.64** |

Table 1: The effect after adding parameter $\beta_0^i$. For AGNews, due to the length limit, we only use three demonstrations per class. Our Anchor Re-weighting method achieves the best performance overall tasks.

from the demonstrations. Given the auto-regressive nature of GPT-like models, where hidden states of tokens depend solely on preceding ones, label words' information aggregation process is independent of subsequent words. This allows for the calculation and caching of the label word hidden states $\boldsymbol{H} = \{\{\boldsymbol{h}_l^i\}_{i=1}^C\}_{l=1}^N$ ($\boldsymbol{h}_l^i$ is the $l$-th layer's hidden state of the $i$-th label word in the demonstration). By concatenating $\boldsymbol{h}_l^1, ..., \boldsymbol{h}_l^C$ at the front in each layer during inference, instead of using the full demonstration, we can speed up inference.

In our preliminary experiments, concatenating hidden states of label words alone was inadequate for completing the ICL task.[5] This might be due to the critical role of formatting information in helping the model to determine the output space at the target position,[6] as highlighted in Min et al. (2022b). As a solution, we amalgamate the hidden states of both the formatting and the label words, a method we've termed **Hidden**$_{\text{anchor}}$.

### 3.2.2 Experiments

We follow the same experimental settings as § 2.2. We compare our Hidden$_{\text{anchor}}$ input compression method with two equally efficient baselines.
**Text**$_{\text{anchor}}$: This method concatenates the formatting and label text with the input, as opposed to concatenating the hidden states at each layer.
**Hidden**$_{\text{random}}$: This approach concatenates the hidden states of formatting and randomly selected non-label words (equal in number to Hidden$_{\text{anchor}}$).
**Hidden**$_{\text{random-top}}$: To establish a stronger baseline, we randomly select 20 sets of non-label words in Hidden$_{\text{random}}$ and report the one with the highest label loyalty.

The **Text**$_{\text{anchor}}$ method is included to demonstrate that the effectiveness of Hidden$_{\text{anchor}}$ is attributed to the aggregation of information in label

| Method | Label Loyalty | Word Loyalty | Acc. |
|---|---|---|---|
| ICL (GPT2-XL) | 100.00 | 100.00 | 51.90 |
| Text$_{\text{anchor}}$ | 51.05 | 36.65 | 38.77 |
| Hidden$_{\text{random}}$ | 48.96 | 5.59 | 39.96 |
| Hidden$_{\text{random-top}}$ | 57.52 | 4.49 | 41.72 |
| Hidden$_{\text{anchor}}$ | **79.47** | **62.17** | **45.04** |
| ICL (GPT-J) | 100.00 | 100.00 | 56.82 |
| Text$_{\text{anchor}}$ | 53.45 | 43.85 | 40.83 |
| Hidden$_{\text{random}}$ | 49.03 | 2.16 | 31.51 |
| Hidden$_{\text{random-top}}$ | 71.10 | 11.36 | 52.34 |
| Hidden$_{\text{anchor}}$ | **89.06** | **75.04** | **55.59** |

Table 2: Results of different compression methods on GPT2-XL and GPT-J (averaged over SST-2, TREC, AG-News, and EmoC). Acc. denotes accuracy. The best results are shown in bold. Our method achieves the best compression performance.

words, rather than the mere text of label words. If we find that Hidden$_{\text{anchor}}$ surpasses Text$_{\text{anchor}}$ in performance, it solidifies the notion that the aggregated information within label words carries significant importance. The **Hidden**$_{\text{random}}$ method is introduced to illustrate that anchor hidden states encapsulate most of the demonstration information among all hidden states.

We assess all compression methods using the label loyalty and word loyalty introduced in § 2.2, in addition to classification accuracy.

### 3.2.3 Results
We can see from Table 2 that the proposed compression method **Hidden**$_{\text{anchor}}$ achieves the best results among all three compression methods on all metrics and for both models. For example, with the GPT-J model, the compression method with anchor states only leads to a 1.5 accuracy drop compared to the uncompressed situation, indicating that the compression introduces negligible information loss. Further, we estimate the efficiency improvements over the original ICL. As shown in Table 3, the speed-up ratio ranges from $1.1\times$ to $2.9\times$, as the efficiency gain is influenced by the length of the demonstrations. We refer readers to Appendix I for

---

[5]Omitting formatting significantly reduces accuracy, as the model will favor common tokens like "the" over label words, indicating confusion about the expected output type.

[6]Here, "formatting" refers to elements like "Review:" and "Sentiment:" in Figure 2.

| Model | SST-2 | TREC | AGNews | EmoC |
|-------|-------|------|--------|------|
| GPT2-XL | 1.1× | 1.5× | 2.5× | 1.4× |
| GPT-J | 1.5× | 2.2× | 2.9× | 1.9× |

Table 3: Acceleration ratios of the Hidden$_{\text{anchor}}$ method.

a more elaborated analysis of the speed-up ratios. Besides, we observe that the acceleration effect is more pronounced in the GPT-J model compared to GPT2-XL, demonstrating its great potential to apply to larger language models.

### 3.3 Anchor Distances for Error Diagnosis

Lastly, we perform an error analysis for ICL by examining the distances between the key vectors in the attention module that correspond to the label words.

#### 3.3.1 Method

Our previous analysis in § 2.3 shows a strong correlation between the model output and $A(q, p_i)$, which is determined by $\mathbf{q}_q \mathbf{k}_{p_i}^T$ as per Eq. 7. Should the key vectors $\mathbf{k}$ for label words $p_i$ and $p_k$ be similar, $A(q, p_i)$ and $A(q, p_k)$ will also likely be similar, leading to potential label confusion. Furthermore, considering the distribution of query vectors $\mathbf{q}_q$, we employ a PCA-like method to extract the components of the key vectors along the directions with significant variations in $\mathbf{q}_q$, denoted as $\hat{\mathbf{k}}$ (see Appendix J for details). We anticipate that the distances between these $\hat{\mathbf{k}}$s can correspond to the category confusion of the model, thus revealing one possible origin of ICL errors. Here, we normalize the distances to a scale of 0-1, with 0 indicating the highest degree of category confusion:

$$\text{Confusion}_{ij}^{\text{pred}} = \frac{\|\hat{\mathbf{k}}_{\mathbf{P_i}} - \hat{\mathbf{k}}_{\mathbf{P_j}}\|}{\max_{s \neq t} \|\hat{\mathbf{k}}_{\mathbf{P_s}} - \hat{\mathbf{k}}_{\mathbf{P_t}}\|}, \quad (11)$$

#### 3.3.2 Experiments

We utilize the GPT2-XL model and TREC dataset, as the model displays varying confusion levels between categories on this dataset. We use all 500 samples of the TREC test set and use 1 demonstration per class for convenience of analysis.

We calculate the actual model confusion score, Confusion$_{ij}$, between category $i$ and category $k$ using the AUC-ROC metric (detailed in Appendix K). We then compare the predicted confusion score, Confusion$_{ij}^{\text{pred}}$, and the actual confusion score, Confusion$_{ij}$, via heatmaps.

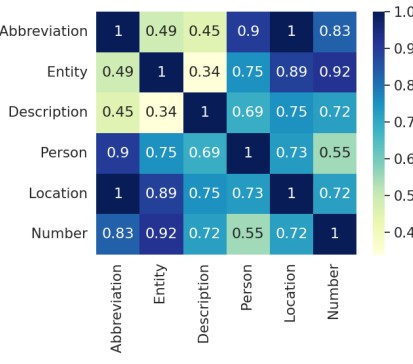

(a) Confusion matrix of Confusion$_{ij}^{\text{pred}}$.

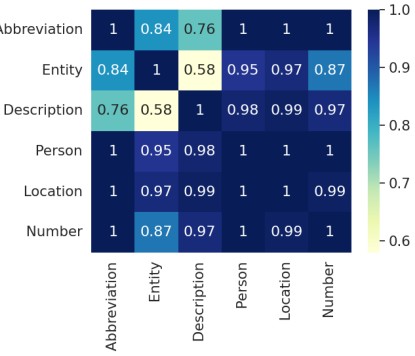

(b) Confusion matrix of Confusion$_{ij}$.

Figure 6: Predicted and real confusion matrix on TREC. We set undefined diagonals to 1 for better visualization. The heatmaps display similarity in confusing category pairs, particularly in lighter-colored blocks.

#### 3.3.3 Results

Figure 6 shows that the proposed approximation metric, Confusion$_{ij}^{\text{pred}}$, can identify the most confusing case (Description-Entity) and performs reasonably well for highly confusing categories (Entity-Abbreviation, Description-Abbreviation). This high correlation indicates that ICL makes errors in categories with similar label anchors. Overall, this result demonstrates that our anchor-based analysis framework could serve as an interpretation tool for better understanding ICL's errors.

### 4 Related Work

The existing literature on in-context learning analysis can be broadly divided into two streams, each focusing on different aspects. The first stream explores the influencing factors of ICL based on input perturbation, such as the order (Min et al., 2022b), the formatting (Yoo et al., 2022; Wei et al., 2022), and the selection of the demonstration (Liu et al., 2022). Designing proper demonstration construc-

tion strategies (Ye et al., 2023; Li et al., 2023a) and calibration techniques (Zhao et al., 2021; Min et al., 2022a) could bring clear boosts to the ICL performance. The second stream investigates the inner working mechanism of ICL through different conceptual lenses, such as making an analogy of ICL to gradient descent (von Oswald et al., 2022; Dai et al., 2022) and viewing the process of ICL as a Bayesian inference (Xie et al., 2022).

In this paper, we provide a novel perspective by examining the information flow in language models to gain an understanding of ICL. Our approach offers new insights and demonstrates the potential for leveraging this understanding to improve the effectiveness, efficiency, and interpretability of ICL.

## 5 Conclusion

In this paper, we propose a hypothesis that label words serve as anchors in in-context learning for aggregating and distributing the task-relevant information flow. Experimental results with attention manipulation and analysis of predictions correlation consolidate the hypothesis holds well in GPT2-XL and GPT-J models. Inspired by the new understanding perspective, we propose three practical applications. First, an anchor re-weighting method is proposed to improve ICL accuracy. Second, we explore a demonstration compression technique to accelerate ICL inference. Lastly, we showcase an analysis framework to diagnose ICL errors on a real-world dataset. These promising applications again verify the hypothesis and open up new directions for future investigations on ICL.

## Limitations

Our study, while providing valuable insights into in-context learning (ICL), has several limitations. Firstly, our research scope was limited to classification tasks and did not delve into the realm of generative tasks. Additionally, our hypothesis was only examined within conventional ICL paradigms, leaving other ICL paradigms such as the chain of thought prompting (CoT) (Wei et al., 2022) unexplored. Secondly, due to hardware constraints, we mainly investigated models up to a scale of 6 billion parameters. Further research that replicates our study using larger-scale models would be beneficial in corroborating our findings and refining the hypotheses set forth in our investigation.

## Acknowledgement

We thank all reviewers for their thoughtful and insightful suggestions. This work is supported in part by a Tencent Research Grant and National Natural Science Foundation of China (No. 62176002). Xu Sun is the corresponding author.

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

Table 4: Demonstration templates and label words. Here <S1> represents the demonstration,  represents the input to be predicted, and <L> represents the label word corresponding to the demonstration. To save space, we only show one demonstration for each task.

| Task | Template | Label Words |
|------|----------|-------------|
| SST-2 | Review: <S1>
Sentiment: <L>
Review: 
Sentiment: | Positive, Negative |
| TREC | Question: <S1>
Answer Type: <L>
Question: 
Answer Type: | Abbreviation, Entity
Description, Person
Location, Number |
| AGNews | Article: <S1>
Answer: <L>
Article: 
Answer: | World, Sports
Business, Technology |
| EmoC | Dialogue: <S1>
Emotion: <L>
Dialogue: 
Emotion: | Others, Happy
Sad, Angry |

Zihao Zhao, Eric Wallace, Shi Feng, Dan Klein, and Sameer Singh. 2021. Calibrate before use: Improving few-shot performance of language models. In *Proceedings of the 38th International Conference on Machine Learning, ICML 2021, 18-24 July 2021, Virtual Event*, volume 139 of *Proceedings of Machine Learning Research*, pages 12697–12706. PMLR.

# Appendix

## A  Experimental Settings

For models, we use GPT2-XL (1.5B) (Radford et al., 2019) and GPT-J (6B) (Wang and Komatsuzaki, 2021) in this paper.

For datasets, we use a sentiment analysis task, Stanford Sentiment Treebank Binary (SST-2) (Socher et al., 2013), a question type classification task, Text REtrieval Conference Question Classification (TREC) (Li and Roth, 2002; Hovy et al., 2001), a topic classification task, AG's news topic classification dataset (AGNews) (Zhang et al., 2015), and an emotion classification task, EmoContext (EmoC) (Chatterjee et al., 2019). The ICL templates of these tasks are shown in Table 4.

## B  Results of $S_{wp}$, $S_{pq}$, and $S_{ww}$ on TREC and EmoC

Figure 7 illustrates the relative sizes of $S_{wp}$, $S_{pq}$, and $S_{ww}$ on TREC and EmoC, mirroring results on SST-2 and AGNews. In shallow layers, $S_{wp}$ (the information flow from the text part to label words)

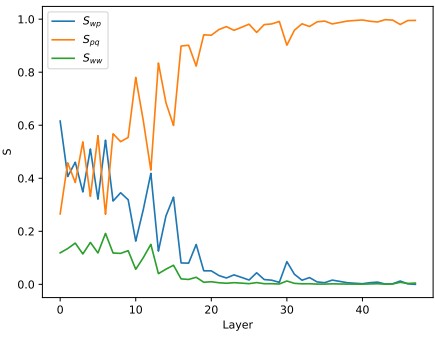

(a) Results on the TREC dataset

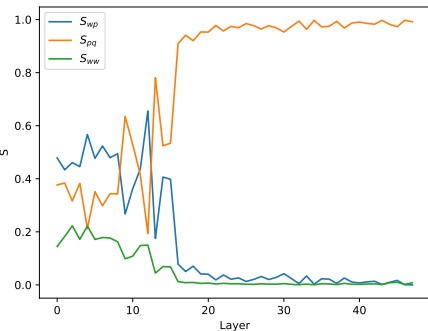

(b) Results on the EmoC dataset

Figure 7: Relative size of $S_{wp}$, $S_{pq}$, and $S_{ww}$ on TREC and EmoC, which is similar to that on SST-2 and AGNews.

is prominent, while $S_{pq}$ (the information flow from label words to targeted positions) is less significant. However, in deeper layers, $S_{pq}$ dominates. Importantly, $S_{wp}$ and $S_{pq}$ generally exceed $S_{ww}$, indicating that interactions involving label words are predominant.

## C  Reason for Using Word Loyalty Besides Label Loyalty

Label loyalty alone may not capture changes in the probability distribution of non-label words or the relative ratio of the probability of the label words within the entire vocabulary. Word loyalty helps address this limitation, which is shown in Table 5.

## D  Isolating Different Numbers of Layers

We study the impact of the numbers of isolated layers, as shown in Figures 8a and 8b. It can be found that isolating shallow layers cause a significant impact, isolating deep layers has a negligible impact on the model, even when the number of isolation layers increases. This further illustrates

| Isolation Layer | Output Label | $V_5$ (sorted by probability) |
|---|---|---|
| First 5 layers | World | "\n", " The", " Google","<\|endoftext\|>", " A" |
| No isolation | World | " World", " Technology", " Politics", " Israel", " Human" |

Table 5: Results on a test sample with the label "World" from AGNews.

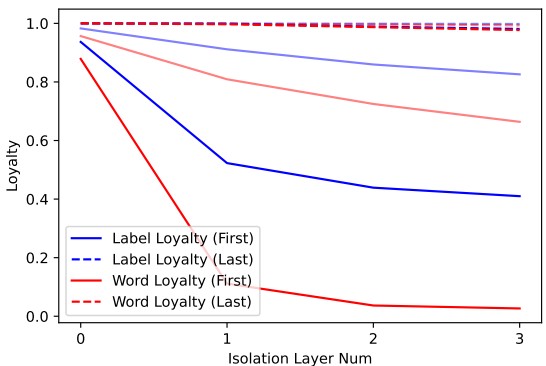

(a) Effect of different numbers of isolated layers on GPT2-XL

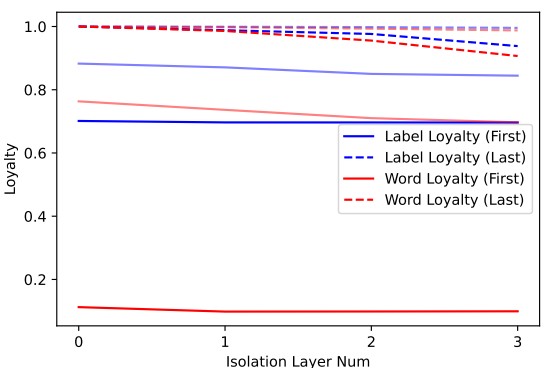

(b) Effect of different numbers of isolated layers on GPT-J

Figure 8: The chart demonstrates variations in label loyalty and word loyalty, dependent on whether label or non-label words are isolated in various layers. 'First' refers to the first several layers, while 'Last' to the last ones. Deep-colored lines represent label word isolation, whereas light colors denote non-label words. Remarkably, isolating label words in the shallow layers significantly influences the outcome, regardless of whether this is compared to isolation in deep layers or to non-label word isolation.

the important role of information aggregation via label words in the shallow layers.

## E Details for the Calculation of AUCROC$_l$

Suppose the positions of the label words in the input $x$ are $p_1, ..., p_C$ (without loss of generality, we suppose $p_i$ corresponds to the $i$th class), the targeted position is $q$, the sum of the attention ma-

trices of all attention heads at the $l$ layer is $A_l$. We postulate that there's a strong correlation between the attention distributions on the label words of the target position $(A_l(q, p_1), ..., A_l(q, p_C))$ and the model's final prediction. We use the AUC-ROC score to quantify this correlation. We regard $(A_l(q, p_1), ..., A_l(q, p_C))$ as a classifier's prediction for the model output label (that is, $A_l(q, p_i)$ is equivalent to the probability of model outputting label $i$), and compute the AUC-ROC value of this prediction relative to the actual model output. We denote this as AUCROC$_l$. For the case with more demonstrations (Appendix F.1), we simply sum up all $A_l(q, p)$ of the same class.

## F Additional Experimental Results

### F.1 Results with More Demonstrations

We implement our experimental analysis utilizing two demonstrations per class, resulting in a total of 4, 12, 8, and 8 demonstrations respectively for SST-2, TREC, AGNews, and EmoC. Our findings, as depicted in Figure 9, Figure 10, and Figure 11, exhibit a high degree of similarity to the results obtained from experiments that employ one demonstration per class.

### F.2 Results for In-Context Learning with semantically-unrelated labels

The applicability of our analytical conclusions to ICL variants, such as the semantically unrelated label ICL (Wei et al., 2023), is an intriguing subject. Given that both GPT2-XL and GPT-J-6B perform at levels akin to random guessing in this ICL setting, we chose LLaMA-33B (Touvron et al., 2023) and SST-2 for our experiment. We substituted labels with 'A'/'B', and adhered to a similar experimental setup as in sections § 2.2 and § 2.3. However, we applied eight shots per class to facilitate the model in achieving an accuracy of 83.0% on SST-2. The outcomes align with those derived in § 2.2 and § 2.3. Figure 12 shows the more pronounced impact of isolating labels in the shallow layers compared to their isolation in the deep layers or the isolation of non-label tokens. Figure 13 con-

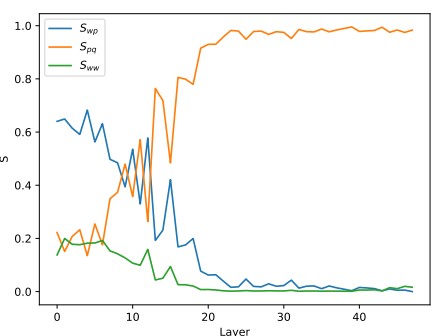

(a) Results on the SST-2 dataset

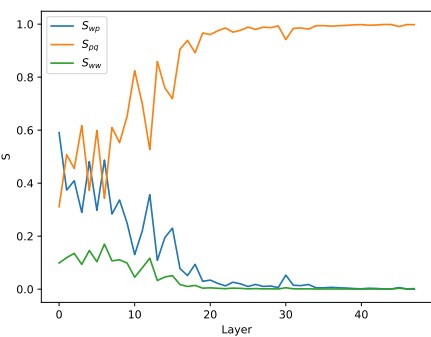

(b) Results on the TREC dataset

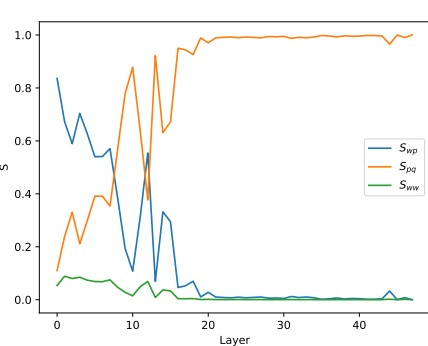

(c) Results on the AGNews dataset

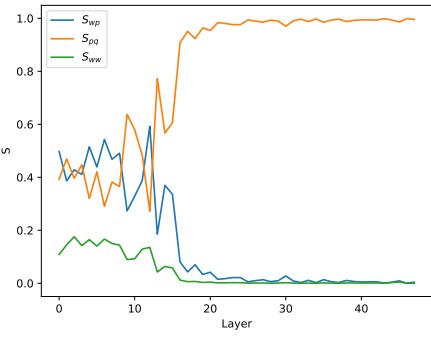

(d) Results on the EmoC dataset

Figure 9: Relative sizes of $S_{wp}$, $S_{pq}$, and $S_{ww}$ when more demonstrations are employed.

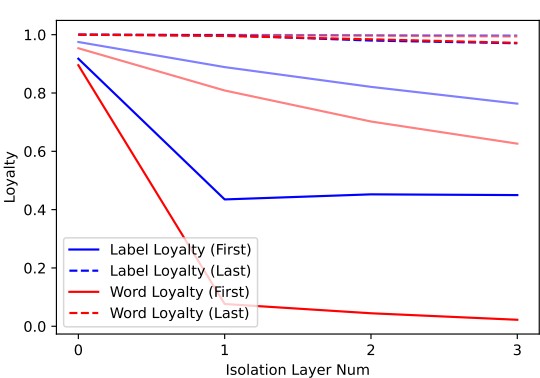

(a) Effect of different numbers of isolated layers on GPT2-XL

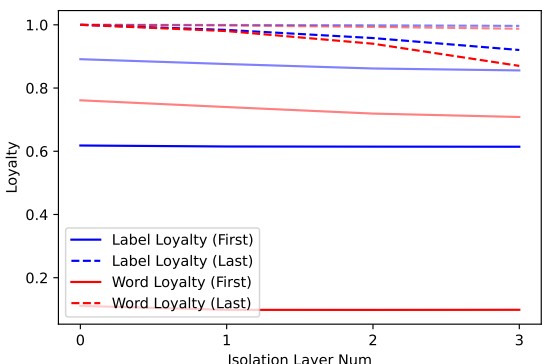

(b) Effect of different numbers of isolated layers on GPT-J

Figure 10: Variations in label loyalty and word loyalty when more demonstrations are employed.

firmed that the model leverages information from anchors in the deeper layers to perform classification.

## G Implementation of Anchor Re-weighting

In order to implement anchor re-weighting, specific adjustments are made in the model's computational process. After calculating the attention matrix $A_l^h$ of the $h$th head in the $l$th layer, we multiply each $A_l^h(q, p_i)$ by $\exp(\beta_{0,lh}^i)$ before proceeding with further computations. This means that for each attention head, we introduce the following modifications:

$$\text{Attention}_l^h(Q, K, V) = \hat{A}_l^h V,$$
$$A_l^h = \text{softmax}\left(\frac{QK^T}{\sqrt{d}}\right),$$
$$\hat{A}_l^h(k, j) = \begin{cases} \exp(\beta_{0,lh}^i) A_l^h(k, j), & \text{if } k = q, j = p_i \\ A_l^h(k, j), & \text{otherwise} \end{cases}.$$

(12)

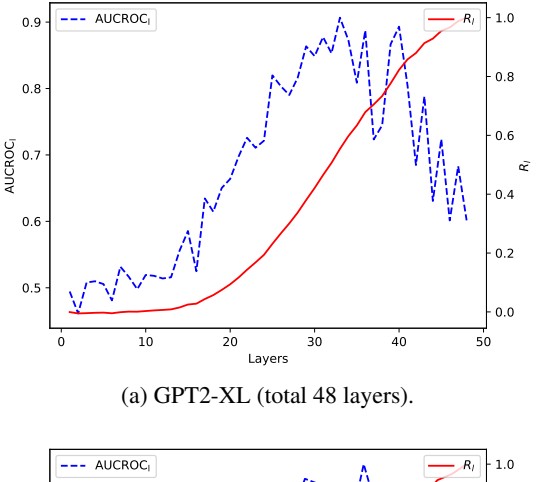

(a) GPT2-XL (total 48 layers).

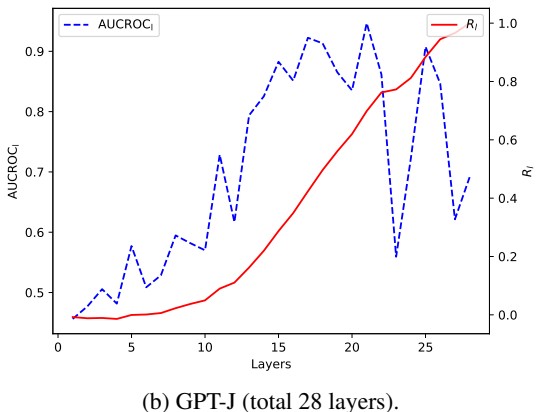

(b) GPT-J (total 28 layers).

Figure 11: $\text{AUCROC}_l$ and $R_l$ of each layer in GPT models when more demonstrations are employed.

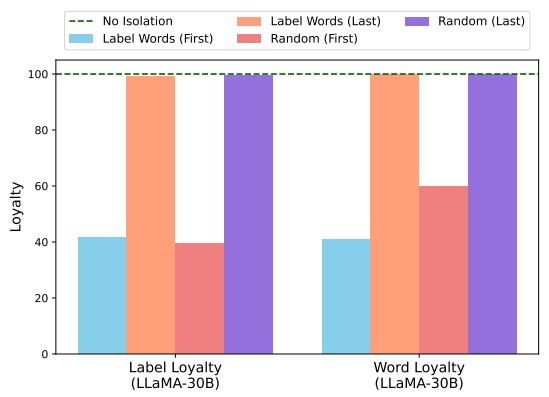

Figure 12: The impact of isolating label words versus randomly isolating non-label words within the first or last 5 layers. Isolating label words within the first 5 layers exerts a more pronounced effect, highlighting the importance of shallow-layer information aggregation via label words.

## H    Training Settings of Anchor Re-weighting

For each random seed, we fix the demonstration and sample 1000 test samples from the test datasets as described in § 2.2. The optimization of parame-

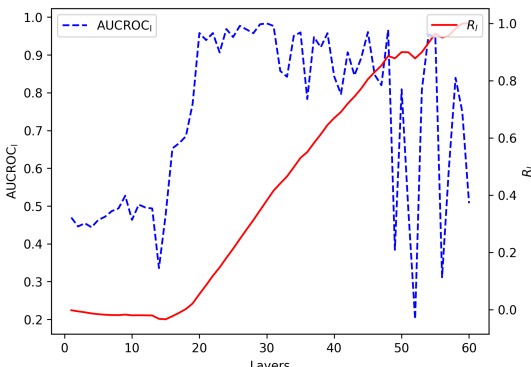

Figure 13: $\text{AUCROC}_l$ and $R_l$ of each layer of LLaMA-33B on SST-2. Still, deep layers display higher relevance to model prediction, reinforcing the idea that the model extracts information from deep-layer anchors for classification.

ter vector $\boldsymbol{\beta}$ is carried out using gradient descent, specifically with the Adam optimizer (Kingma and Ba, 2015). The learning rate is set at 0.01, with $\beta_1 = 0.9$ and $\beta_2 = 0.999$. Due to memory constraints, we use a batch size of 1. This optimization process is repeated for 10 epochs. Owing to limitations in computational resources, we restrict our evaluation to the GPT2-XL model and exclude the GPT-J model from our assessment.

## I    The Factor of $L_{\text{demo}}$ and $L_{\mathbf{x}}$

|  | SST-2 | TREC | AGNews | EmoC |
|---|---|---|---|---|
| GPT2-XL | 1.1× | 1.5× | 2.5× | 1.4× |
| GPT-J | 1.5× | 2.2× | 2.9× | 1.9× |
| $L_{\text{demo}}$ | 18 | 61 | 151 | 53 |
| $L_{\mathbf{x}}$ | 19 | 7 | 37 | 12 |

Table 6: Acceleration ratios, $L_{\text{demo}}$ and $L_{\mathbf{x}}$.

From Table 6, we observe a correlation between the acceleration ratios and the ratio of the total demonstration length ($L_{\text{demo}}$) to the length of the text predicted ($L_{\mathbf{x}}$). It suggests that a greater ratio of total length to predicted text length may yield a higher acceleration ratio.

In addition, the table illustrates that datasets with longer demonstration lengths tend to exhibit higher acceleration ratios. For instance, the AGNews dataset, which has the longest $L_{\text{demo}}$, presents the highest acceleration ratio among the datasets analyzed. These findings could indicate an increased efficiency of the Hidden$_{\text{anchor}}$ method in contexts involving longer demonstration lengths.

## J  Calculation of $\hat{\mathbf{k}}$

For the sampled sequence $x_1, ..., x_T$ to be predicted, we denote the query vectors of the target positions as $\mathbf{q}_1, ..., \mathbf{q}_T$. We then compute the matrix $\hat{\mathbf{Q}} = (\mathbf{q}_1 - \overline{\mathbf{q}}, ..., \mathbf{q}_T - \overline{\mathbf{q}})$ by subtracting the mean vector, $\overline{\mathbf{q}}$, from each query vector. Subsequently, we determine the $M$ directions, $\mathbf{v}_1, ..., \mathbf{v}_M$, that correspond to the M largest variation directions for the centralized query vectors $\hat{\mathbf{q}}_1, ..., \hat{\mathbf{q}}_T$. The $i^{th}$ direction, $\mathbf{v}_i$, is chosen to maximize the variance of the projection of the centralized query vectors onto it, while also being orthogonal to the previously chosen directions, $\mathbf{v}_1, ..., \mathbf{v}_{i-1}$. This process can be formalized as follows:

$$
\begin{aligned}
\mathbf{v}_1 &= \underset{\|\mathbf{v}\|=1}{\arg\max} \, \mathrm{Var}\left\{\mathbf{v}^\top \hat{\mathbf{Q}}\right\}, \\
\mathbf{v}_2 &= \underset{\|\mathbf{v}\|=1, \mathbf{v}\perp\mathbf{v}_1}{\arg\max} \, \mathrm{Var}\left\{\mathbf{v}^\top \hat{\mathbf{Q}}\right\}, \\
&\cdots \\
\mathbf{v}_M &= \underset{\|\mathbf{v}\|=1, \mathbf{v}\perp\mathbf{v}_1,...,\mathbf{v}\perp\mathbf{v}_{M-1}}{\arg\max} \, \mathrm{Var}\left\{\mathbf{v}^\top \hat{\mathbf{Q}}\right\}.
\end{aligned}
\tag{13}
$$

We define $\sigma_i$ as the square root of the variance of the projection of $\hat{\mathbf{Q}}$ onto the $i^{th}$ direction, i.e., $\sqrt{\mathrm{Var}\left\{\mathbf{v}_i^\top \hat{\mathbf{Q}}\right\}}$.

To derive features $\hat{\mathbf{k}}$s, we project the key vector $\mathbf{k}$ onto the directions $\mathbf{v}_1, ..., \mathbf{v}_M$ and scale the projections by the corresponding standard deviations $\sigma_1, ..., \sigma_M$. Each feature, $\hat{\mathbf{k}}_i$, is thus calculated as $\sigma_i \mathbf{v}_i^T \mathbf{k}$.

We further examine the influence of $M$ on the prediction confusion matrix, Confusion$ij^{\mathrm{pred}}$, as depicted in Figure 14. Given the similarity in outcomes for various $M$, we settle on a value of $M = 10$ for computation of Confusion$ij^{\mathrm{pred}}$.

## K  Calculation of Confusion$_{ij}$

To gauge the true degree of confusion between categories $i$ and $k$ for a given model, we suggest utilizing the Confusion$_{ij}$ metric:

First, we procure all test samples $x_t$ bearing true labels $i$ or $k$. We then obtain the probabilities $p_i^t$ and $p_j^t$ yielded by the model for categories $i$ and $k$, respectively, on these samples. These probabilities are normalized to a total of 1. Essentially, we derive a classifier $f$ that delivers the probabilities $p_i^t$ and $p_j^t$ for the categories $i$ and $k$ respectively, on the test samples $x_t$. By calculating the Area Under the Receiver Operating Characteristic Curve (AUC-ROC) value of this classifier $f$, we get the degree of confusion between category $i$ and $k$, termed as Confusion$_{ij}$.

The computed Confusion$ij$ is a value that never exceeds 1. The closer Confusion$ij$ approximates 1, the less pronounced the confusion, and vice versa.

We use the above metric instead of directly analyzing the output labels of the model because previous work has indicated the issue of insufficient output probability calibration in ICL (Zhao et al., 2021), which is greatly affected by factors such as sample ordering and model preferences for specific label words. By leveraging our defined degree of confusion, Confusion$_{ij}$, we can implicitly alleviate the disturbances arising from insufficient probability calibration on the output labels. This allows for a more accurate representation of the model's degree of confusion for different categories, mitigating the impact of randomness.

## L  Reproducibility

In the supplementary material, we have provided codes that allow for the faithful replication of our experiments and subsequent result analysis. To ensure consistency and reproducibility across different devices, we have fixed the five random seeds to the values of 42, 43, 44, 45, and 46. We invite readers to delve into the code for additional implementation details that may arouse their interest.

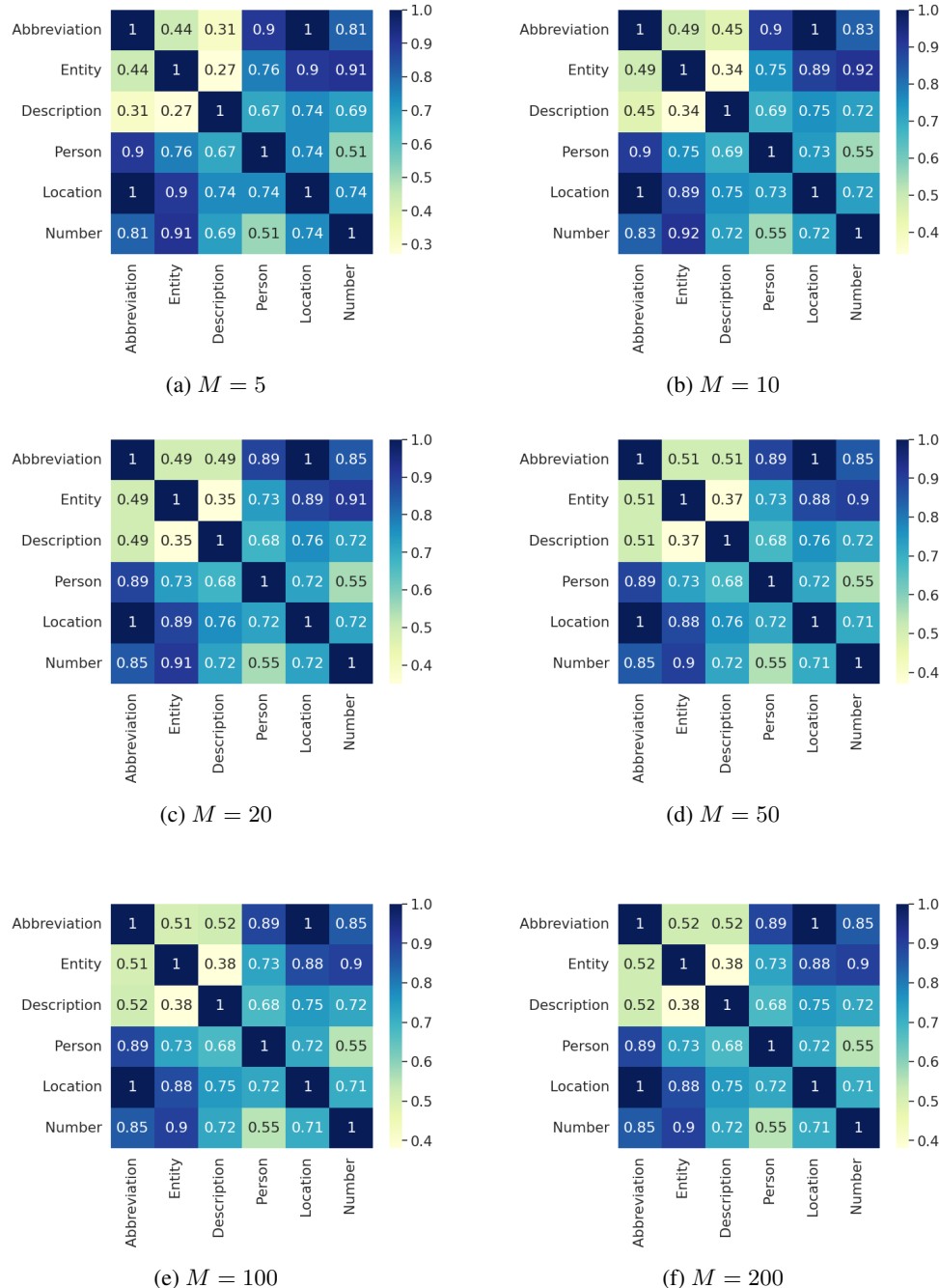

Figure 14: Predicted confusion matrices under $M = 5, 10, 20, 50, 100, 200$.