# OpenReview forum: "Label Words are Anchors: An Information Flow Perspective for Understanding In-Context Learning"
_EMNLP/2023/Conference — EMNLP 2023 Main_

### Official Review · Reviewer_ghKz · 2023-07-27

**Soundness:** 5

**Excitement:**

4: Strong: This paper deepens the understanding of some phenomenon or lowers the barriers to an existing research direction.

**Paper Topic And Main Contributions:**

This paper explores the underlying mechanism of how in-context learning works for classification tasks.

Through the saliency score of the attention matrix between tokens (context tokens, label words, target position token) in shallow and deep layers, they observe that label words function as anchors.

Particularly, in the shallow layers information aggregates into label word representations and in the deep layers information of label words largely contribute to the label prediction of the input.

Three experiments are designed to leverage this observation: 1) re-weighting the attention of the label words to the target position by a few learnable parameters; 2) compressing each demonstration into only label words; 3) utilizing the distance between key vectors of the label words to predict the confusion matrix. These experiments show performance improvement thus verify the correctness of the observation.

**Questions For The Authors:**

1. Both the "Label words" and "label tokens" are used across the paper. Are they with the same meaning? Which token will the information flow in and gather from if the label word contains more than one tokens (subwords). The answer to this question will help extending the observation to generation tasks, where the label usually contains more than one tokens.

**Reasons To Accept:**

How ICL works is an important question and this paper provides insights of it on a specific setting, the classification task.

The correctness of the observation is verified through thorough experiments.

Although the observation is only on the classification task where the label words contain few (1?) tokens, it is possible to extend to a more general situation including generation task. This work encourages relevant research on interpretable LLM.

**Reasons To Reject:**

Refer to question 1.

**Reproducibility:**

4: Could mostly reproduce the results, but there may be some variation because of sample variance or minor variations in their interpretation of the protocol or method.

**Reviewer Confidence:**

3: Pretty sure, but there's a chance I missed something. Although I have a good feel for this area in general, I did not carefully check the paper's details, e.g., the math, experimental design, or novelty.

---

> ### Author Rebuttal · Authors · 2023-08-28
>
> **Question 1:** Both the "Label words" and "label tokens" are used across the paper. Are they with the same meaning?
>
> **Answer 1:** Sorry for causing confusion. In our experimental settings, ‘label words’ are roughly equal to ‘label tokens’. The only exception is the ‘Abbreviation’ in the TREC dataset, which has more than one subword. For this exception, we simply use the first subword in our paper, as Calibrate Before Use: Improving Few-Shot Performance of Language Models (Zhao et al., 2021) does.
>
> **Question 2:** Which token will the information flow in and gather from if the label word contains more than one token (subword)?
>
> **Answer 2:** We explore the importance of the subwords {‘Ab’, ‘be’,’vation’} in ‘Abbreviation’ via saliency metrics (we use the sum of all saliency scores corresponding to a specific subword as its importance). In this case, the relative importance of {‘Ab’, ‘be’,’vation’} is about 4: 1: 3.5. This suggests that, at least in this instance, the first subword holds the most significance, followed by the last subword, with the middle word being the least important.
>
> **Question 3:** What about generation tasks?
>
> **Answer 3:** For generation tasks, the current token-level anchor might need to be expanded to span-level ones (or the weighted combination of multiple tokens), and their function might differ due to the difference between classification and generation. We look forward to exploring this in the future.

---

### Official Review · Reviewer_XFQJ · 2023-08-03

**Soundness:** 4

**Excitement:**

4: Strong: This paper deepens the understanding of some phenomenon or lowers the barriers to an existing research direction.

**Paper Topic And Main Contributions:**

Paper topic and main contributions: This paper introduces an analysis perspective by using label words as anchors to understand ICL among different layers. This analysis derives two facts: (1) in shallow layers, label words gather information from demonstrations; (2) in deep layers, label words leads to the final prediction. Based on these facts, the authors propose three methods (anchor re-weighting, anchor-only context compression and anchor distances for error diagnosis) improve the effectiveness, efficiency, and interpretability of ICL.

**Reasons To Accept:**

1. Clear analysis of the information flow with label words as anchors to explain the ICL mechanism.
2. Proposed anchor re-weighting is simple and effective to make significant improvements over vanilla ICL.
3. Proposed anchor-only context compression has faster inference speed than the original ICL while remain comparable performance.

**Reasons To Reject:**

1. More analysis should be done since recent studies explore the influence of different ICL formats on the final prediction, such as random labels, reversed labels (e.g.,True->False, False->True), label agencies (replace labels with meaningless characters) and so on. I think the label words anchors analysis on these setups will make deeper insights and more robust conclusions. The current analysis is clear but seems to be trivial.
2. I note that anchor re-weighting uses auxiliary training set, which seems to be a unfair comparison with the vanilla ICL, since one core advantage of ICL is no extra fine-tuning when transferred to new tasks. For a fair comparison, I think fine-tuning the model on the auxiliary training set is a necessary baseline.
3. In anchor-only context compression, for hidden_random, I think a more persuasive choice is to select non-label words that show significant influence on label words in shallow layers. Since previous analysis has demonstrated the information aggregation. The current random non-label words selections also seems to be too trivial and inadequate.
4. I am confused about the advantages of using anchor distances for error diagnosis as no other tools provided to detect similar label words.

**Reproducibility:**

4: Could mostly reproduce the results, but there may be some variation because of sample variance or minor variations in their interpretation of the protocol or method.

**Reviewer Confidence:**

3: Pretty sure, but there's a chance I missed something. Although I have a good feel for this area in general, I did not carefully check the paper's details, e.g., the math, experimental design, or novelty.

---

> ### Author Rebuttal · Authors · 2023-08-28
>
> **Question 1:** Analysis of different ICL formats on the final prediction like random labels, reversed labels (e.g.,True->False, False->True), and label agencies (replace labels with meaningless characters) may be helpful.
>
> **Answer 1:**
>
> In experiments, we find that GPT2-XL and GPT-J-6B perform similarly to random guessing in these different ICL formats, so we do not analyze them. Even for llama-30b, we find that only label-agency ICL works. We then conducted label-agency ICL with llama-30b on SST-2 (in this case, the model can achieve an accuracy of 0.83). **The results are similar to those in Sections 2.2 and 2.3 in our paper, thereby reinforcing our initial conclusions.**
>
> The detailed results are listed below:
>
> 1. The graph of AUCROC_{l} and R_l is similar to Figure 5 in the paper. For all 60 layers of llama-30b, AUCROC_{l} is about 0.5 for layers 0-20, about 0.9 for layers 20-50, and fluctuates for layers 50-60 as in Figure 5.
> 2. The results for isolating label words in the first and last layers are similar to Section 2.2. Isolating labels in the first several layers has a greater impact than isolating in the last several layers or isolating non-labels. There is only one minor difference: label loyalty of isolating labels in the first several layers is slightly higher than that of isolating non-labels in the first several layers.
>
>    |               | Word Loyalty | Label Loyalty |
>    | ---------------------------- | ------------ | ------------- |
>    | Isolating labels (first)   | 40.8     | 41.7     |
>    | Isolating labels (last)   | 100     | 99.0     |
>    | Isolating non-labels (first) | 60.0     | 39.3     |
>    | Isolating non-labels (last) | 100     | 99.3    |
>
> Given the consistency of these results with our earlier findings, we can conclude that our observations and insights remain valid. We hope this detailed response addresses your concerns.
>
> **Question 2:** I note that anchor re-weighting uses an auxiliary training set. So can you add fine-tuning the model on the auxiliary training set as a baseline to justify the superiority of anchor re-weighting?
>
> **Answer 2:** Here, our primary intention behind proposing anchor-reweighting was to validate our anchor hypothesis and **show the significance of anchors in ICL**. We do not aim to claim it as a new state-of-the-art technique in a much border range. We have tested finetuning, and the result is 83.2 (SST-2), 67.6 (TREC), 81.2 (AGNews), and 55.7 (EmoC), lower than anchor-reweighting in SST-2 and AGNews, but higher in TREC, EmoC, and the average. However, it is worth noting that finetuning trains all parameters of the model. In contrast, the number of parameters trained in our anchor re-weighting is **less than 0.002% of the total model parameters**. Our anchor re-weighting method is more parameter-efficient while maintaining competitive performance.
>
> **Question 3:** In anchor-only context compression, for hidden_random, can you develop a stronger baseline by selecting non-label words with significant influence?
>
> **Answer 3:** To decide which combination of non-label words from demonstrations has the most significant influence, a direct way is to check the loyalty metric of all possible combinations of non-label words, but it is too time-consuming and thus computationally intractable. As a practical compromise, we devised a surrogate approach. We randomly generated 20 combinations of non-label words and subsequently selected the one exhibiting the highest label loyalty among them. The results for this approach (denoted as **Hidden_{top-20}**) are shown below (averaged over SST-2, AGNews, TREC, EmoC). **This approach's performance is better than Hidden_{random} while still worse than Hidden_{anchor}, which further validates our conclusion:**
>
> | GPT2-XL     | Word Loyalty | Label loyalty | Accuracy |
> | --------------- | ------------ | ------------- | -------- |
> | ICL (No compression) | 100     | 100      | 51.90  |
> | Hidden_{anchor} | 62.17    | 79.47     | 45.04  |
> | Hidden_{random} | 5.59     | 48.96     | 39.96  |
> | Hidden_{top-20} | 4.49     | 57.52     | 41.72  |
>
> | GPT-J-6B    | Word Loyalty | Label loyalty | Accuracy |
> | --------------- | ------------ | ------------- | -------- |
> | ICL (No compression) | 100     | 100      | 56.82  |
> | Hidden_{anchor} | 75.04    | 89.06     | 55.59  |
> | Hidden_{random} | 2.26     | 49.03     | 31.51  |
> | Hidden_{top-20} | 11.36    | 71.10     | 52.34  |
>
> **Question 4:** I am confused about the advantages of using anchor distances for error diagnosis. There might be other tools for error diagnosis.
>
> **Answer 4:**  Here, our primary intention behind introducing the 'Anchor Distances for Error Diagnosis' approach was to **investigate and illustrate the potential correlation between anchor distances and model misclassification.** This method offers a novel perspective to interpret model misclassification through the lens of anchor distances. For now, we use it mainly as **an interpretation method**. We will try to explore the possibility of developing it into a mature tool for error diagnosis in the future.

---

### Official Review · Reviewer_5cZV · 2023-08-05

**Typos Grammar Style And Presentation Improvements:** 1. "Demonstration tokens in the demon…
**Soundness:** 5

**Excitement:**

5: Transformative: This paper is likely to change its subfield or computational linguistics broadly. It should be considered for a best paper award. This paper changes the current understanding of some phenomenon, shows a widely held practice to be erroneous in someway, enables a promising direction of research for a (broad or narrow) topic, or creates an exciting new technique.

**Paper Topic And Main Contributions:**

This paper analyses in-context learning and reveal that the label words in the demonstration examples function as anchors where semantic information is aggregated into the label word representations during initial shallow layers and the information from label words serve as reference for LLM's predictions in deep layers. This is well supported by experiments. The enhancements to ICL proposed are great.

**Questions For The Authors:**

A. I am curious how the information flow from the test-time text to the target position across layers looks like? This would indicate the information contribution of actual text w.r.t. demonstrations.
B. Similarly, how would the information flow into test-time text tokens look like?
C. Anchor reweighting looks very much similar to adapter in principle. Can you expand on the similarities?
D. In lines 410-416, I didn't quite understand why formatting is important? Please explain it
E. I didn't understand Text_{anchor} in 3.2.2 experiment.

**Reasons To Accept:**

- Great analysis
- Neatly presented
- The further enhancements of ICL proposed that are inspired by the analysis would have broader impact.

**Reasons To Reject:**

I must admit that I liked the whole paper and I am being very stringent while mentioning the following points.

1. In figure 5, when we look into the rate of increase in cumulative function, it was the highest after layer 10 for GPT J and layer 20 for GPT2-XL. From this, we can conclude that, not just the deepest layers such as (40-50) contribute, even the middle ones such as layer 20-25 also contribute well. This shows that anything after the initial layers start extracting information from label words. Maybe this would require a partial rewording of the usage "deep" layers in the hypothesis-2 or mentioning a number for what "deep" indicates should be sufficient.

**Reproducibility:**

5: Could easily reproduce the results.

**Reviewer Confidence:**

4: Quite sure. I tried to check the important points carefully. It's unlikely, though conceivable, that I missed something that should affect my ratings.

---

> ### Author Rebuttal · Authors · 2023-08-28
>
> ### **For questions:**
>
> **Question(A):** How does the information flow from the test-time text to the target position across layers look like? This would indicate the information contribution of actual text compared to demonstrations.
>
> **Answer(A):**
> To picture the flow, we resorted to saliency scores. We utilized two metrics to assess the nature of the flow:
>
> 1. **Coefficient of Variation (std/mean)**: We calculate the ratio std/mean of saliency scores corresponding to the test-time text tokens within a specific layer. This illuminates the variance in the influence of different tokens.
> 2. **Relative influence of layers**: We compare the saliency scores of different layers. This indicates the importance of this flow in different layers.
>
> Our results are listed below:
>
> - For the **Coefficient of Variation**, as layers go deeper, the disparity in token influence tends to widen. Generally speaking, the initial layers (approximately the first 15 layers among the total 48 layers in GPT2-XL) show a modest coefficient of variation (around 1). In contrast, this value increases in subsequent layers (to about 1.5). However, the results may differ across tasks. For instance, AGNews exhibits a consistently high coefficient of variation (higher than other datasets by about 0.5).
> - For **Relative influence of layers**,  the influence is small for the initial layers, then peaks around the 20th-30th layer, and decreases in the last several layers.
>
> Additionally, the relative contribution of the test-time text v.s. demonstrations were quantified via saliency. They are 0.65:1 (SST2), 3.46:1 (TREC), 2.39:1 (AGNews) and 2.09:1 (EmoC). So the relative contribution may be task-dependent.
>
>
> **Question(B):** How does the information flow into test-time text tokens look like?
>
> **Answer(B):**
> We use the metrics outlined in Answer(A) to picture the flow. The result is:
>
> 1. The **Coefficient of Variation** for the flow is generally above 2, higher than that in Question(A).
> 2. As for the **Relative influence of layers**, the initial 20 layers seem to be dominant, and there's a pronounced decline in influence in subsequent layers.
>
> **For (A) and (B):** In brief, the dynamics and characteristics of the information flow about test-time text are intricate. We look forward to delving deeper into this realm in future research.
>
> **Question(C):** Anchor re-weighting's similarity to adapters
>
> **Answer(C):** Actually, anchor reweighting can be seen as a special type of adapter, since it introduces a small number of parameters and keeps most parts of the model as original, but it is designed based on our anchor hypothesis and owns fewer parameters than normal adapters. We will include a discussion in the related work section in the revised version.
>
> **Question(D):** Why formatting is important as discussed in lines 410-416
>
> **Answer(D):** As emphasized in Rethinking the role of demonstrations: What makes in-context learning work (Min et al., 2022b), formatting plays an important role in ICL. **Formatting ensures that the model is oriented correctly to perform ICL** (or the subsequent compression). In experiments, when formatting is omitted, the accuracy significantly drops, and the most likely tokens predicted by the language model tend to be frequently occurring tokens like ‘the’, rather than the label words. This phenomenon indicates that the model is confused when formatting is omitted and does not know what kind of output is expected.
>
> **Question(E):** Explain the reason for introducing Text_{anchor} in 3.2.2 experiment.
>
> **Answer(E):** The primary aim behind introducing Text_{anchor} is to show that Hidden_{anchor} works due to the aggregated information in label words, rather than the pure text of label words. When we observed that Hidden_{anchor} outperforms Text_{anchor}, it reinforces the idea that the aggregated information in label words is important.
>
> **About the term ‘Deep':** Thanks for pointing out this. We will clarify the meaning of 'deep' and mention a number for what "deep" indicates in the revision.
>
> **For typos and presentation suggestions:** Thanks for your kind suggestions. We will revise our paper based on your suggestions 1, 2, and 3.

---

### Meta-Review · Area_Chair_E6A5 · 2023-09-15

**Recommendation:** 5

**Metareview:**

This paper provides a deep dive into the mechanics of in-context learning (ICL) for classification tasks by analyzing how label words function as anchors.

In general, the paper is well-structured and written. Proposed anchor re-weighting and anchor-only context compression that improved the effectiveness and inference speed over vanilla ICL. All reviewer agreed the paper could have a wider impact and encourages relevant research on interpretable LLM.

Some concerns were also pointed out by several reviewers that require additional experiments and clarification.

---

### Decision · Program_Chairs · 2023-10-07

**Decision:**

Accept-Main

**Comment:**

This paper provides a deep dive into the mechanics of in-context learning (ICL) for classification tasks by analyzing how label words function as anchors.

In general, the paper is well-structured and written. Proposed anchor re-weighting and anchor-only context compression that improved the effectiveness and inference speed over vanilla ICL. All reviewer agreed the paper could have a wider impact and encourages relevant research on interpretable LLM.

Some concerns were also pointed out by several reviewers that require additional experiments and clarification.